

# Long Range Plasma Momentum Coupling by High Voltage Static Electric field and Deep Space Exploration

Kokwei Chew[1], Xinyu Zhou[1], and Yian Lei[1]

[1]School of Physics, Peking University, Beijing 100871, China

*Correspondence to*: Yian Lei (yalei@pku.edu.cn)

**Abstract**. Space exploration has been long constrained by the efficiency and capability of modern chemical rocket. Propellantless propulsion has been proposed as a solution to expand the boundary of space exploration. In this paper, we examine the possibilities of a propellantless propulsion scheme through the interaction between the spacecraft and ambient plasma. The spacecraft is charged to high electric potential by constantly shooting electrons away. The high voltage spacecraft will deplete the surrounding electrons, thus interact with a wide range of the background plasma (solar wind) and thus effectively extract momentum from the plasma. By taking advantage of the exploitable ambient plasma, a spacecraft can reach very high speed, thus considerably reducing the travel time. The scheme is also applicable for braking, which is helpful in the exploration of inner planets like Venus and Mercury, and the stopping at the destination planets or stars.

## 1 Introduction

Space exploration is the next step of human development. The main difficulty experienced in space travel is the long distance across planets, stars, or galaxies, and the low limitation of fuel. The average distance between planets in solar system is about 5.5 Astronomical units (AU) or 800 million km away, and the closest star system, the Alpha Centauri is 4.37 light-years ($4.1 \times 10^{16}$ m) away from solar system. Contemporary spacecraft can hardly reach a speed of 40 kms$^{-1}$, with the help of multiple slingshot acceleration of the planets. The speed is far too low, considering the travelling to the Alpha Centauri will takes longer than the entire human history.

Propellantless propulsion is a scheme proposed to overcome the problems of limited fuels carried by chemical rockets as well as ion thrusters. To date, various kinds of propellantless schemes such as electric sail and solar sail has been proposed as an alternative to chemical rockets for interplanetary or interstellar space travelling. In this paper, we propose a propellantless propulsion scheme by coupling the spacecraft with the ambient plasma to extract its momentum. The spacecraft is charged to high electric potential by constantly shooting electrons far away. The high voltage spacecraft will interact with a wide range of the background plasma (solar wind) in momentum and gain thrust. Comparing with the electric sail proposed by P. Janhunen (2007, 2014), the effective radius of the spacecraft can be held within 100 m to provide better control and manoeuvring.

The key challenge in the engineering for this scheme is maintaining high potential of the spacecraft. The spacecraft has to accelerate and shot electrons far away to remain high potential and couple with ambient plasma. As electron is the lightest charged particle, with a mass three orders of magnitudes smaller than the lightest ion, hence can be easily deflected by



magnets, electron acceleration is the easiest and most durable in engineering. Conventional non-superconducting electron
injectors (Bluem et al. 2004) can accelerate electrons to 5 MeV with a compact size, and operate continuously at a maximum
current of 1 A. These two parameters set an upper limit on the potential and current of our scheme.
The feasibility of maintaining high potential also limited by the field ion evaporation effect or field evaporation (FEV) (Zurlev
et al. 2003). FEV is the removal of surface atom from its own lattice structure by induced electric field. The whole process is
rather complicated and not well studied yet. In general, when the electric field of a surface exceeded certain threshold, called
$E^e$, there will be significant increase in the FEV process and deplete the structure. A theoretical estimation given by Dinko N.
Zurlev (Forbes, 2006) showed that the $E^e$ for most metals is at the order of 10 Vnm$^{-1}$. For our spacecraft design with an
effective radius of 100m, the threshold potential is at the order of $10^{12}$ V, which is 6 orders of magnitude beyond our limit and
thus should not be a problem.
**2 Concept**
The spacecraft interact with the ambient plasma through static electric field. Comparing with the electric sail design as
proposed by P. Janhunen, our concept consists of a spacecraft located at the centre and linked to several positively charged
lightweight balls by conducting tethers. The distance from spacecraft to the balls could be a few hundreds of meters. The
spherical shape of balls is designed to reduce the FEV effect, and the distance of the balls is to effectively lower the potential
of the structure, hence lower the energy of the ejected electron beam and improve energy efficiency. Due to electric static
repulsion; all the balls will distributed uniform around the spacecraft. When deployed for operation, the electron gun is pointed
to the opposite or perpendicular direction of the solar wind for ejecting the electrons away from the spacecraft.

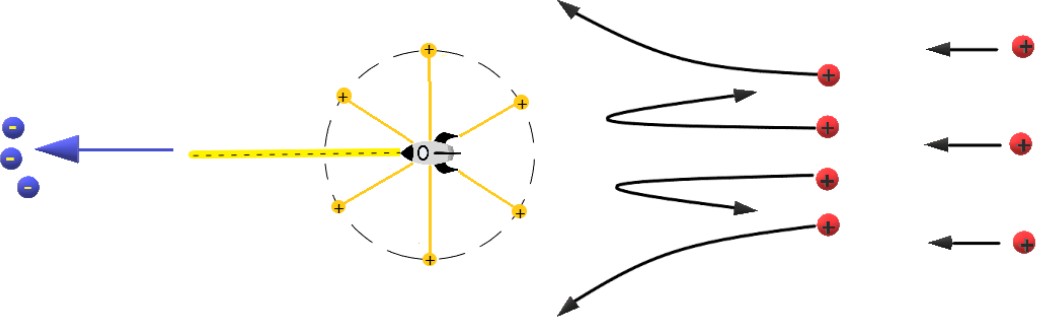


**Figure 1: plasma wind coming from the right side being deflection by the electrostatic field of the spacecraft. The whole structure is**
**consisted of a main unit at the center and connected to charged balls attached to the main unit.**
For the sake of simplicity in simulation, the whole structure is treated as a sphere with the distance from spacecraft
to the balls as the radius of the imaginary sphere. The sphere is charged to a surface potential of $V_0$, A stream of plasma (solar
wind) moving from the positive x direction. We consider only 1 species of ion, the protons. The deflection of protons from its
trajectories provides the momentum. The loss in x-direction momentum of protons is transferred to the spacecraft. To calculate



the trajectory and hence the momentum, the potential generated by the sphere need to be addressed. For theoretical estimation,
we consider the potential to be a central force and takes the forms of either Coulomb potential or Debye potential.

58        In the absence of plasma, the potential generated by the sphere would be a simple Coulomb potential:

$V(r) = \frac{V_0 r_0}{r},$ (1)
Where $r_0$ is the radius of the sphere and $V_0$ is the potential at the surface of the sphere and the potential at far point is zero.
Due to the presence of ambient plasma, shielding occurred and the potential dropped by and exponential factor. The effective
potential is:
$V(r) = \frac{V_0 r_0}{r} e^{-\frac{r-r_0}{\lambda_D}},$ (2)
Where $\lambda_D$ is the effective Debye length, $\lambda_D = \sqrt{\varepsilon_0 (V_0 - V_1)/n_0 e}$ (Janhunen et al. 2016). $V_1$ is the kinetic energy of the
incoming proton.

66        We can estimate thrust by solving the equation of motion of the proton in a Coulomb potential. The deflection or

scattering of protons, $\theta$ from its original direction is calculated from:
$b = \frac{V_0 r_0 e}{2 E_k} \cot\frac{\theta}{2},$ (3)
Where $b$ is the impact parameter, $E_k$ is the kinetic energy of protons in solar wind. In this case, we set $\theta$ to 90 degree, the
impact parameter $b$ represent how far the electric field can affect and repel the protons from solar wind. Assuming $V_0 = 1$ MV,
$r_0 = 100$ m, $E_k = 1.5$ keV,  we can have the impact parameter $b = 33$ km and thus an effective scattering cross section of
3400 km². The solar wind pressure at 1 AU from the Sun is 2 nPa and the spacecraft is estimated to acquire 6.8 N of thrust at
the Earth's orbit.

74        In reality, the potential can never be a symmetric Coulomb potential due to the presence of plasma. The next part of

this paper will provide simulation result for the actual thrust obtained by the spacecraft.
**3 1D Simulation**
In 1D simulation, the system will reach a static condition after the spacecraft is charged. The particles (ions) interact with the
charged spacecraft and eventually the potential around the spacecraft takes the form as described by equation 2. The deflection
of protons from its own trajectory contribute the x-direction momentum to the spacecraft. As shown in figure 2, the spacecraft
is located at the origin. The ions is coming from the +x direction and assumed to have initial velocity in the x-direction. The
motion of the particle calculated by Runge-Kutta method with adaptive time-steps for error control.





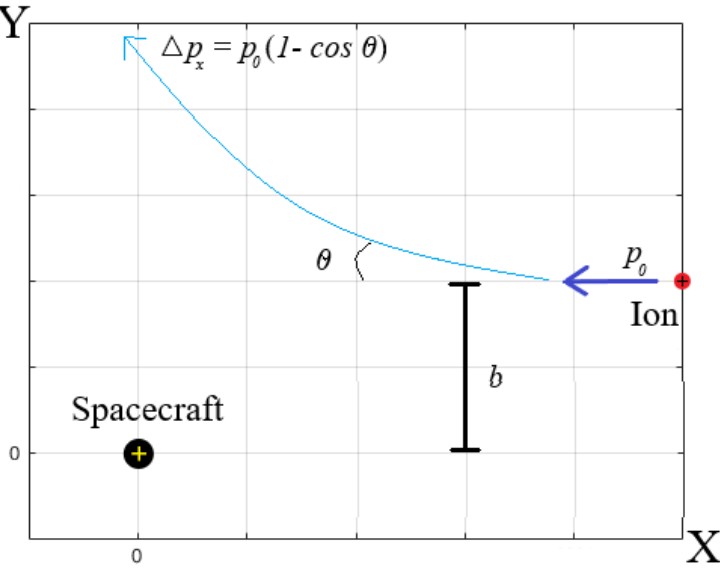

**Figure 2: The spacecraft located at the origin and the ions from solar wind coming from the positive x direction.**

The model for 1D simulation is rather simple. It provides the relation between the thrust generated and key parameters like spacecraft surface potential $V_0$, radius $r_0$ and solar wind density $n_0$. This provide a basis for calculating the trajectories for interplanetary travel for our spacecraft. Figure 3 and table 1 shows the scaling relation between the thrust and surface potential of surface as well as solar wind density.

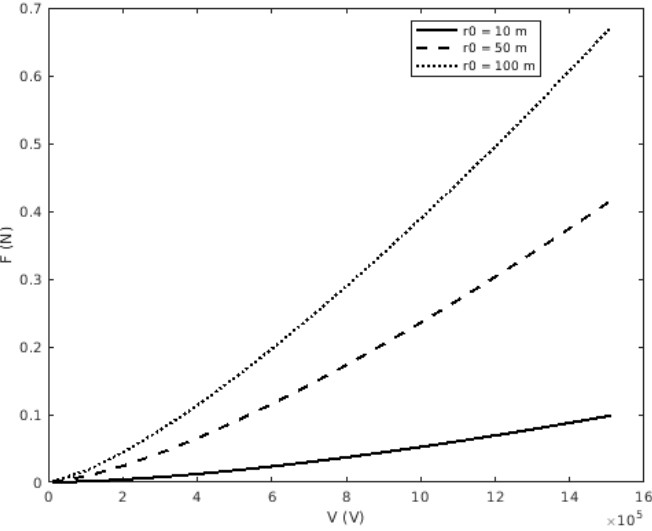

**Figure 3(a): The relation between thrust $F$ and the surface potential of the sphere $V_0$. Dotted line: $r_0 = 100$ m, dashed line: $r_0 = 50$ m, solid line: $r_0 = 10$ m. Plasma density $n_0 = 1.2 \times 10^6$ m$^{-3}$.**






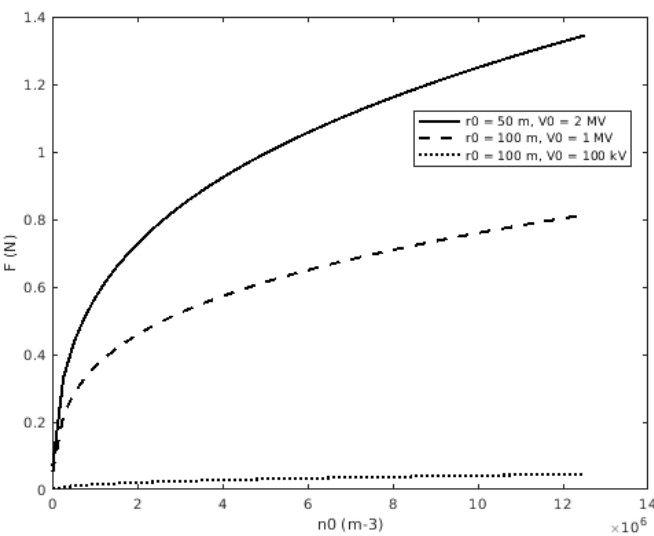


**Figure 3(b): Figure 3(b) The relation between thrust $F$ and the plasma density of solar wind $n_0$. Dotted: $r_0 = 50$ m, $V_0 = 2$ MV,**
**dashed: $r_0 = 100$ m, $V_0 = 1$ MV, solid: $r_0 = 100$ m, $V_0 = 100$ kV.**
**Table1: scaling relation between thrust $F$ and $V_0$, $n_0$, $r_0$.**

| $n_0$/ $10^5$ m$^{-3}$ | $V_0$/ V | $r_0$/ m | Scaling |
|---|---|---|---|
| 5.0 | - | 10 | $F \propto V_0^{1.4715}$ |
| 5.0 | - | 50 | $F \propto V_0^{1.3257}$ |
| 5.0 | - | 100 | $F \propto V_0^{1.2771}$ |
| 1.0 | - | 100 | $F \propto V_0^{1.3290}$ |
| - | 100 k | 10 | $F \propto n_0^{0.5723}$ |
| - | 100 k | 100 | $F \propto n_0^{0.4360}$ |
| - | 1.0 M | 10 | $F \propto n_0^{0.5330}$ |
| - | 1.0 M | 50 | $F \propto n_0^{0.3724}$ |
| - | 1.0 M | 100 | $F \propto n_0^{0.3183}$ |
| 5.0 | 100 k | - | $F \propto r_0^{0.8603}$ |
| 1.0 | 1.0 M | - | $F \propto r_0^{0.6804}$ |
| 5.0 | 1.0 M | - | $F \propto r_0^{0.6804}$ |


To estimate the electrons flux, we use the Orbital Motion Limited (OML) theory [1][10]. An electron will be captured
when its trajectory meets the surface of the sphere. The electrons from solar wind coming from 1 direction, if the impact
parameter (distance of electron from centre of sphere and perpendicular to its own velocity) lies within certain value, the





electron will be absorbed by the sphere. Assuming that the electrons only come from solar wind, and experience a central force.
With simple mechanic invariants, we can calculate the current:
$I = \pi r_s{}^2 e n_0 v,$ (4)
Where $n_0$ is the density of the electron in solar wind, $v$ is the velocity of incoming electron. $r_s = r_0\sqrt{1 + \frac{2eV_0}{mv^2}}$ is the impact
parameter. All electrons aimed at impact parameter less than $r_s$ will hit on the surface of the sphere.

## 4 2D Simulation

In this section, we provide a PIC 2D simulation describe the behavior of the particles. The simulation region is has a cylindrical
symmetry along the x-axis.  Similar to figure 2, both the protons and electrons constantly flow in from the +x direction. The
electric field is solved by Coulomb equation. The potential of the sphere (spacecraft) is $V_0$. All the electrons moving inside the
sphere are absorbed.
The simulation span across a region of 18 km * 9 km, with spatial grid varying linearly from 10 m to 1800 m. The
grid size is small at the origin to provide a more precise result around the spacecraft. With plasma density of $n_0 = 5.0 \times$
$10^4 \text{ m}^{-3}$, the Langmuir frequency for electron is equivalent to $\omega_{pe} = 1.3 \times 10^4 \text{ s}^{-1}$. The timestep is set to $\Delta t = 7.75 \times$
$10^{-5}$ s. We run 2 cases, (1) $V_0 = 1 \text{ MV}, \ r_0 = 100 \text{ m}$ , (2) $V_0 = 2 \text{ MV}, \ r_0 = 50 \text{ m}$, with $T = 32000\, \Delta t$.
Figure 4 shows the established stable contour plot of the potential. We can see there is a high potential area on the
incoming side of the solar wind. This means the spacecraft can couple or interact with a large range of the ambient plasma.
Figure 4(b) is for ion density. The concentration of ions on the right side is because the ions have nowhere to go. Figure 4(c)
is for electron density; note the high concentration of electrons around the spacecraft.





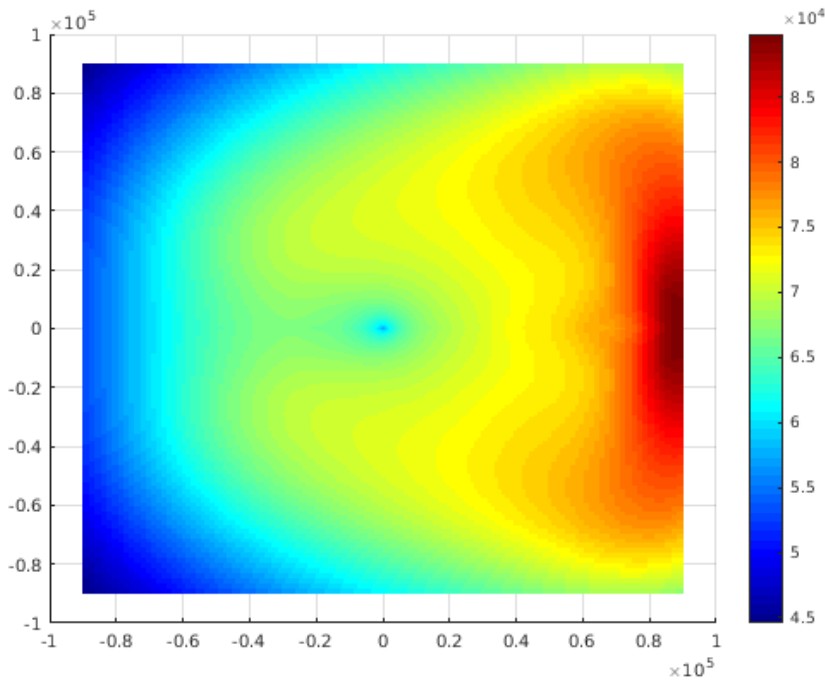


**Figure 4(a): The plasma potential when simulation reached static state ($T = 32000\ \Delta t$, $\Delta t = 7.75 \times 10^{-5}$ s), the spacecraft located**
**at the origin.**

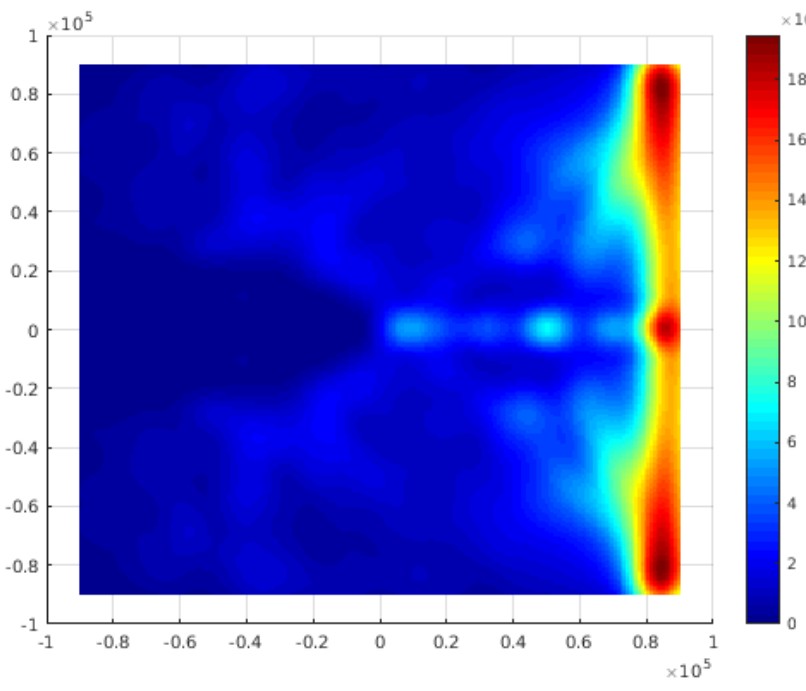




**Figure 4(b): The ion density when simulation reached static state. Ions is repelled by the sphere, hence low ion density at the left**
**side. A large portion of ions accumulation at the right side of the region is due to the boundary effect.**

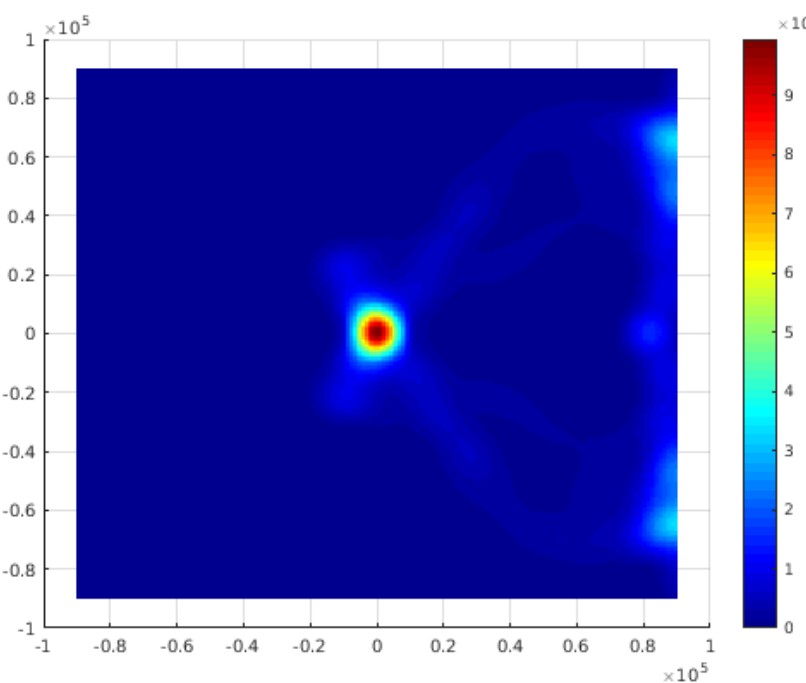


**Figure 4(b): The electron density when simulation reached static state. Electrons in the region almost depleted.**

127       The thrust acting on the sphere and the electron current is calculated and compared with 1D simulation in table 2. In

2D simulation, the thrust is 2~3 times larger than that of 1D simulation, and the electron current lower. Hence, the momentum
coupling between the spacecraft and the ambient plasma is stronger than plasma shielding.

**Table 2: Comparison between 1D and 2D simulation result.**

| $V_0$,$r_0$ | 1D Simulation | | 2D Simulation | |
|---|---|---|---|---|
| | Thrust / N | Current / mA | Thrust / N | Current / mA |
| 1 MV,100 m | 0.52 | 168 | 1.6 | 102 |
| 2 MV,50 m | 0.77 | 84 | 1.6 | 73 |

**5 Interplanetary Travel**
In this section, we consider a simplified journey starting from Earth's orbit to outer planet such as Mars and Jupiter. A
comparison between result from 1D and 2D simulation showed that the 1D still a good approximation for thrust calculation.




The thrust generated depends on the ambient plasma density $n_0$, and $n_0$ drops by inversely square law of the distance, $D$, from
the Sun.
Assuming
$F(D) = A \cdot D^k,$         (5)
Where F is the thrust, D is the distance to the Sun. $k$ is a scaling coefficient obtained from 1D simulation. $A$ is a
constant determined by 2D simulation. From 1D simulation, $k$ is within the range from -1.3 to -1.5. Adding the force into the
equation of motion of the spacecraft:
$m\left(\ddot{D} - D\dot{\theta}^2\right) = -\frac{GM_Sm}{D^2} + F(D),$         (6)
Where $M_S$ and $m$ are the mass of the Sun and spacecraft respectively, $\theta$ is the angular coordinate of the spacecraft in
a polar coordinate system with the Sun located at the center. By solving the equation of motion numerically with initial distance
being the radius of the orbit of the Earth, radial velocity being zero and tangential velocity equal to that of Earth, we can
calculate the trajectories of the spacecraft. Table 3 shows the time taken for each mission under various circumstances. The
main power consumption is from the electron accelerator (gun).

Table 3: Time taken for spacecraft to a journey to reach different planets.

| $V_0$ | $r_0$ | $m$ | Initial acceleration | Time taken (days) | | | | Power |
|---|---|---|---|---|---|---|---|---|
| (MV) | (m) | (kg) | (mms$^{-2}$) | Mars | Jupiter | Saturn | Pluto | (kW) |
| 0.1 | 100 | 59 | 0.85 | 267 | - | - | - | 11 |
| 1 | 100 | 500 | 3.64 | 80 | 290 | 446 | 1055 | 102 |
| 1 | 100 | 1000 | 1.82 | 120 | 503 | 774 | 1758 | 102 |
| 2 | 50 | 500 | 4.02 | 76 | 277 | 430 | 1039 | 73 |
| 2 | 50 | 1000 | 2.01 | 114 | 471 | 732 | 1717 | 73 |


The most efficient way for sending spacecraft to a target planet is by orbit transfer [11]. The spacecraft is transfer
from a smaller orbit (Earth) to a larger orbit (Mars) following a Hohmann transfer orbit. In our case, for planets like the Mars,
the spacecraft can follow a similar orbit for energy efficiency. With a surface potential $V_0 = 1$ MV, effective radius $r_0 = 100$ m,
a spacecraft of mass $m = 460$ kg can be transferred to the Mars.
**6 Result and discussions**



We proposed an ambient plasma momentum coupling spacecraft propelling scheme by utilizing high electric potential for the
spacecraft to interact with a wide range of background plasma, which can be considered as a compact electric sail with a much
smaller structure.
Preliminary calculations show that it is promising in space exploration. There are some discrepancies between the 1D
and 2D simulation but generally, the order of the thrust and electron current are the same. The 2D simulation result shows that
with a surface potential of the spacecraft of 1 MV, this scheme can produce a thrust of about 1.6 N, by consuming a power of
about 100 kW. The thrust-energy efficiency is comparable to contemporary plasma thrusters (Mason et al. 2001, Richard 2004).
The simulation is performed in low plasma density. In reality, this scheme could generate a larger thrust than our calculation.
Since this scheme requires no propellant, it has an advantage over chemical rockets and ion thrusters. Comparing with the E-
sail proposed by P. Janhunen, our spacecraft require a relatively small structure (100 m versus 10 km). It is much easier to
control or manoeuver. A simple calculation shows that a spacecraft of 500 kg can reach Mars in 80 days and Pluto in 3 years.
The scheme is also applicable in braking, as long as the momentum of ambient plasma is exploitable. In situations
like braking near Jupiter, travelling to the inner planets, the plasma trapped by Jupiter and the solar wind can be used for
braking.
This scheme can achieve very high interstellar travelling speed by delivering artificial dense and energetic beams to
the spacecraft over a very long distance, by, for example, a series of powerful particle accelerators on the Moon, other satellites,
or dwarf planets, thus drastically shorten the travelling time to the nearest stars from tens of thousands of years to a few
hundreds of years.
If the situation is favorable, such as in a cosmic jet, a spacecraft could be accelerated to relativistic velocities.

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
