# Peer review of "Long Range Plasma Momentum Coupling by High Voltage Static Electric field and Deep Space Exploration"

_Annales Geophysicae, 2019_

## Referee Comment (RC1) · Anonymous Referee #1 · 23 May 2019

The authors proposed a concept of propellantless propulsion scheme, a spacecraft linked to several positively charged lightweight balls by conducting tethers. The scheme has a much smaller structure size than the E-sail and can operate with a very high potential to achieve a large thrust by interacting with exploitable ambient plasma to considerably reducing the travel time of a spacecraft.

This scheme is very well designed. It could be more practical and important if the authors can tell more about the size and the potential material of the lightweight balls and why these balls can decrease the structure size to two orders smaller than the large E-sail. In addition, the authors showed several cases in Table 3. The power consumptions are very large (73-103kW) except the first case. The large power consumption may not be suitable for the spacecraft nowadays. I think the author should discuss how the spacecraft power meets the power consumption of their propellantless propulsion, or they can add more cases in Table 3 for the journey to Jupiter, Saturn and Pluto with a power consumption similar to the first case.

other comments and technical corrections:

Line 45-46: 'the distance of the balls is to effectively lower the potential of the structure'. What is the structure, the spacecraft only or the whole scheme? Please explain why 'the distance of the balls' can effectively lower the potential?

Line 115: Which of the two cases is shown in Figure 4? Which direction is the X-axis in Figure 4? Please add labels and units in the figure.

Line 129: Please explain why 'the momentum coupling between the spacecraft and the ambient plasma is stronger than plasma shielding'.

Line 134: the thrust in 2D simulation is 2∼3 times larger than that of 1D simulation. So, it doesn't seem like 'a good approximation'.

Line 140: k ranges from -2+0.3=-1.7 to -2+0.5=-1.5. right?

Line 150: what does the '[11]' mean?

Line 161: What is the thrust-energy efficiency of this scheme proposed by the authors?

---

## Author Comment (AC1) · 27 May 2019

The authors are grateful to the careful reading and valuable comments from the referee.

We have revised the manuscript according to the comments. Here is our reply:

1, about the power consumption and structure size. Yes the power consumption is a unrealistic, we changed the size of the spacecraft, and lowered the power. Actually we can also increase the effective radius to, say, 1 km, thus lower the power to about one tenth of the orginal. We have some problem with the numerical stabilities of the simulation. Janhunen's orginal E-Sail interact with neutral plasma, in which the influence is limited by Debye length. We create a large dynamic positively charge cloud, which can interact with a much larger range of ambient plasma. If E-Sail can deplete the neibouring electrons, it should be as efficient too.

2, the effective radius of a conductor. With the same amount of electric charge, the larger ths conductor is, the lower its potential will be. The energy required to shot the electrons away (to keep the potential of the spacecraft) should be a little higher than the potential. Lower potential is prefered for energy efficiency.

3, Line 115: Which of the two cases is shown in Figure 4? Which direction is the X-axis in Figure 4? Please add labels and units in the figure. Please find in the modified manuscript attached.

4, Line 129: Please explain why 'the momentum coupling between the spacecraft and the ambient plasma is stronger than plasma shielding'. This is due to the difference of quasi-neutral plasma, and non-neutral structure as explaned in comment no.1 above.

Line 134: the thrust in 2D simulation is 2âĹij3 times larger than that of 1D simulation. So, it doesn't seem like 'a good approximation'. Yes, you are right, we changed the manuscript.

Line 140: k ranges from -2+0.3=-1.7 to -2+0.5=-1.5. right? The number is -1.3 to -1.5.

Line 150: what does the '[11]' mean? We have changed to the right reference style.

Line 161: What is the thrust-energy efficiency of this scheme proposed by the authors? This is a preliminary proposal, and we don't have experimental support. We believe the efficency should be better than we presented, for we also tried to solve the electro-magnetic field numerically, but we cannot get a stable result, though seems to be much better. We tried two years, and fail to reach a presentable simulation. Both students (first two authors) graduated, and the work is not supported, so we have to settle with the conservative but referencible method and result. The general principle applies: the larger the effective radius, the higher the energy efficiency.

The manuscript has been revised and highlighted.

Please also note the supplement to this comment:
https://www.ann-geophys-discuss.net/angeo-2019-41/angeo-2019-41-AC1-supplement.pdf

**Supplement:**

[revised manuscript text omitted]

---

## Referee Comment (RC2) · Anonymous Referee #2 · 3 Jul 2019

General comments

The authors suggest an "improved" electric sail design which is more compact than the original tether-based electric sail and thus could provide potential benefit in terms of manoeuverablity. Unfortunately, it seems to me that the power budget estimations are not correct, and even if they were correct, the power requirement is so high that the concept is hardly competitive against the original electric sail (with moderate voltage, as originally proposed, or perhaps with increased voltage: if we are ready to increase voltage to MV in this concept, the same could be done with the original concept as well) and electric propulsion. Because of these fundamental problems, I cannot recommend

the paper for publication.

Specific comments

1) By Eq. (4) I get much larger currents than those in Table 2. If I assume 10 eV electron temperature in the solar wind so that v=1.8e6 m/s, I get 7 kA current. This would lead to 7 gigawatts of power consumption at 1 MV potential.

2) One could obtain a estimate without simulation by simply solving at which value of r the V(r) of Eq. (2) has the value V_1 and calling this r the maximum impact parameter. I am not sure but the result might even be expressable analytically in terms of Lambert's W function.

3) I think it would be sensible to compare this system with the ordinary E-sail geometry where V0 has the same value as here. The present design is more compact, but I would guess that the ordinary electric sail model has lower power versus thrust ratio, because of the mathematical forms of the thrust and OML current expressions for cylindrical and spherical cases. The cylindrical case yields less electrons being collected, per unit of thrust produced, because the attracted electron flux converges in only one dimension rather than two.

4) Even if the power estimate of 1e5 W would be right, it corresponds to very high power versus thrust ratio, several orders of magnitude higher than the electric sail.

5) At line 101, it is not true that all electrons arrive from the solar wind direction, because in the solar wind, the electron thermal speed is usually larger (by factor of four or five, say) than the bulk speed. Hence electrons arrive from all directions. Probably this does not change the form of equation (4), though.

6) The spheres are at ends of tethers. If the tethers are bare, there will be enormous electric field on their surface, much larger than on the surface of the sphere. If they are insulated, the insulator must be relatively thick, perhaps ∼1 cm, in order to withstand 1 MV voltage. Such insulation might add considerable weight.

---

## Author Comment (AC2) · 14 Jul 2019

The authors are grateful to the referee for the careful reading and insightful comment. We hope we can have better discussion with the referee about his/her concerns. Here is our reply:

About the general comments:

Our concept differs with the original electric sail design in the following ways: 1. Size and maneuverability. 2. The physical model of the way the spacecraft interact with the background plasma / solar wind. 3. The range/quantity of the plasma the spacecraft can interact with, hence the amount of beneficial momentum the spacecraft can harvest. 4. The tethered balls are designed to effectively lower the potential of the whole structure, hence improve the power efficiency. We are concerned with the original design with the numerical/physical model of the way the spacecraft/tether interact with the plasma. We think a global dynamic electro-magnetic field simulation is more realistic, because the depletion of electron around the spacecraft would yield a global effect. That is what we had been doing in the past few years, but we experience numerical problems and cannot have a stable and presentable result. We can share some facts of the simulation here: the power efficiency is much better, in at least one order of magnitude; as the simulation scale is limited by about tens of kilometers, the field was always significantly affected by the boundary conditions; the magnitude difference between the vicinity of the spacecraft and the far away regions is always a problem; we found the force the spacecraft experienced is oscillating, even in an absolutely stable solar wind, which is understandable, because of Langmuir oscillation, and this is only achievable by a realistic field solver. The original idea of long thin lines has the following difficulty except for the size: 1. It is not realistic to maintain a constant voltage on a thin and long conducting line, let alone in a conducting environment (plasma). 2. The assumption that the current only comes from the single particle trajectory collision of the electrons is not well grounded. 3. If we put charge in a conductor, the charge automatically distributes to the tipping points of the conductor, as we dealt with in our paper. Of cause, our concept is still a preliminary proposal, and some number in the original paper does not look good, such as the power assumption, but we also pointed out the power efficiency can be improved by increasing the effective radius, which still have a large room. Actually we have modified this part in response to the first referee. One important reason for our seeking of publication is: This work is not officially supported, as the students graduated, the concept will simply die. We also hope the concept could attract attention of experimental scientists, because proof of concept experiment is simple, just blow a charged ball with a plasma. The ideas of braking, artificial particle beam momentum delivering, cosmic jet surfing, etc., are still

worth sharing.

As for the specific concerns: 1. Not all electron come from the solar wind will be intercepted by the structure. In our case, the effective radius is different from the collisional radius of the electron trajectory, because the effectiveness is for the potential, not for interception of the electron. The space is still very much empty. 2. Yes we could explore more in this direction. One of our concerns is that it is hard to set a cut-off limit for the potential, for it is now more likes a Coulomb one. 3. Please see the general explanation. 4. The effective radius we used in the original paper is a little too small. We have modified the manuscript with a larger effective radius and better power efficiency in responce to the first referee. 5. We do use a simpler assumption that all particles comes from the incoming direction of the solar wind with the same speed (hence no temperature). This could affect the single particle model, but eventually, only the potential, the current, and the distribution of the potential matter. 6. We have discussed this in the last paragraph of the introduction, which seems to be not a problem.